# Development and Reliability of the Oxford Meat Frequency Questionnaire

**DOI:** 10.3390/nu13030922

**Published:** 2021-03-12

**Authors:** Cristina Stewart, Kerstin Frie, Carmen Piernas, Susan A. Jebb

**Affiliations:** Nuffield Department of Primary Care Health Sciences, University of Oxford, Radcliffe Observatory Quarter, Woodstock Road, Oxford OX2 6GG, UK; kerstinfrie@gmail.com (K.F.); carmen.piernas-sanchez@phc.ox.ac.uk (C.P.); susan.jebb@phc.ox.ac.uk (S.A.J.)

**Keywords:** dietary assessment, meat consumption, meat intake, food frequency questionnaire, meat frequency questionnaire

## Abstract

Reliable and valid measurements of meat intake are needed to advance understanding of its health effects and to evaluate interventions to reduce meat consumption. Here, we describe the development and reliability of the Oxford Meat Frequency Questionnaire (MFQ). It asks individuals to report the number of servings of meat and seafood products they consumed in the last 24 h and is administered daily over 7 days. The MFQ combines food portion size data from the UK Food Standards Agency with estimates of meat content in composite dishes from the UK’s National Diet and Nutrition Survey. Adults who self-reported to eat meat (*n* = 129) completed a 3-week web-based test–retest reliability study assessing the MFQ twice, with a wash-out week in the middle. Two-way random intraclass correlation coefficients (ICC) revealed moderate to good agreement on all meat outcomes (total meat ICC = 0.716; minimum–maximum individual components = 0.531–0.680), except for fish and seafood (ICC = 0.257). Participants reported finding the questionnaire easy to use and, on average, completed it in less than 2 min. This new MFQ offers a quick, acceptable, and reliable method to measure changes in an individual’s meat intake in a UK population.

## 1. Introduction

Dietary intake assessments are essential for studying the associations between diet and health-related diseases as well as measuring the impact of dietary interventions. Food diaries, dietary recalls and food frequency questionnaires (FFQs) are amongst the most frequently used dietary assessment methods [1]. Food diaries are a prospective dietary assessment method in which a respondent records all of the food and beverages they consume over a specific period of time, most commonly three or seven days. Although they can be a precise form of dietary measurement, they carry a high level of participant and investigator burden. Dietary recalls offer an alternative with a lower participant burden–they are open-ended interview-like surveys asking participants about their food and beverage consumption over a pre-specified time frame. However, they require a trained interviewer and come with a considerable analysis effort for the investigator. FFQs are popular in the literature because they provide a method to efficiently measure dietary intake with low participant and low investigator burden. This type of questionnaire asks participants how often and how much food they have eaten over a specific period. Whilst the advantages of FFQs are considerable, it is important to note that, like dietary recalls, they are prone to recall bias [1,2].

A variety of FFQs have been developed and validated for the UK, which capture the whole diet of a participant [3], such as the UK Biobank WebQ [4]. To make them as efficient to complete as possible, these FFQs often group food items at higher levels, thus making it difficult to assess the consumption of specific food categories at a granular level. Specialized FFQs, focusing on a defined set of food and drink items provide an opportunity to measure intake more accurately for specific categories of interest [3,5,6].

Research on meat consumption has received increased attention in recent years, showing clear associations between the intake of red and processed meat and adverse health outcomes [7]. Meat products have significant environmental impact and there are calls to reduce meat production and consumption to help achieve carbon net zero targets [8]. To further advance research in this area, researchers need to be able to collect precise measurements of meat intake and measure changes in intake over time. To the best of our knowledge, only two meat frequency questionnaires have been developed and reported so far. The first was developed in North America and focusses on meat types which may contain heterocyclic amines, a chemical associated with the development of cancer [9]. The second questionnaire, which was developed for a Norwegian population, focused on seafood intake only [10]. Neither questionnaire provides a holistic assessment of meat intake from all sources. Moreover, neither of them have been tested in a UK setting, where dietary habits and portion sizes may differ. As a result, previous evaluations of these questionnaires may not be transferable [11].

The aim of this research was therefore to develop a meat frequency questionnaire to estimate changes in meat intake in a UK sample of adults. This paper presents the development of the Oxford Meat Frequency Questionnaire (MFQ) and the results of a test–retest reliability assessment.

## 2. Materials and Methods

### 2.1. Development of the MFQ

#### Objectives

The first aim was to create a questionnaire which could be employed to measure changes in meat intake, for instance in intervention or observational studies. In this context, we define meat intake as the intake of any animal flesh, including fish and seafood. Meat intake may vary considerably on a day-to-day basis, for example with individuals following flexitarian diets or taking part in the ‘Meat Free Monday’ campaign, and hence the questionnaire needs to cover several days to accurately reflect usual meat intake. To minimize recall bias, we aimed to create a 24 h MFQ which can be administered daily over 7 days. This is similar to the method successfully employed in the UK Biobank WebQ–which measures an individual’s whole diet through 24 h FFQs, administered four times across the year [4].

Our second aim was to measure absolute meat consumption with a precision which would make it possible to measure small changes in consumption, for example, resulting from portion size reduction. To reduce participant burden, we decided to ask respondents to indicate the number of half servings consumed of a given product, and used underlying portion size and meat content data from the literature to estimate how much meat they had.

Different sources of meat may have different health effects. To reflect this, our final aim was to separately measure intake for the following animal groups: beef, pork, chicken and turkey, lamb and mutton, game, and fish and seafood. The questionnaire was split into these categories, allowing us to calculate separate mean intakes for each animal group, as well as for red and processed meat.

### 2.2. Development Process

The development process was five-fold (see Appendix A for more detail):

(i) We used The UK Food Standards Agency (FSA) food portion sizes book to identify typical meat products and extract their mean portion size information. Portion sizes were exclusive of bone or shell where possible [12].

(ii) We categorized all items into MFQ categories. All meat items were first grouped according to animal source (e.g., beef, chicken and turkey, pork, fish and seafood), then commonly consumed product types (e.g., mixed dish, cuts, slices, pies and pastries) and finally portion size in grams. For instance, the chicken cordon bleu was added to the ‘chicken and turkey animal’ group and sorted together with other ‘coated products’ of a similar average portion weight. ‘Mixed dishes’ encompassed composite dishes containing both meat and non-meat ingredients (e.g., curries, casseroles and stews).

(iii) We extracted information on the meat content of composite dishes from the food composition database of the Year 10 UK’s National Diet and Nutrition Survey (NDNS), which closely reflected items extracted from the FSA food portion sizes book [13]. Products containing more than one type of meat (e.g., chicken and prawn paella) were assigned to the animal group with the highest proportion of meat, with quantities of all meat types summed together. For the mixed dishes within each animal group, we sorted the NDNS products by meat content and split them into low, medium and high meat proportions aiming to have near to equal representation in all groups. With this, mixed dishes containing <25% meat were classed as ‘low meat’, dishes containing 25–40% meat were classed as ‘moderate meat’, and dishes containing >40% meat were classed as ‘high meat’. These groups were treated as separate MFQ categories. We then calculated the mean grams of meat per serving for each MFQ category.

(iv) We calculated mean portion sizes for each MFQ category based on FSA portion size data to provide a single serving size. The mean meat content was first calculated for each FSA item, and then across all items in an MFQ category. To calculate the grams of meat per serving for each MFQ category, the mean serving size for the category was multiplied with its respective mean meat content. Figure 1 shows an excerpt of the extracted data for the chicken and turkey category. The MFQ and its underlying meat proportions are available in Appendix A.

(v) Finally, we created serving size indicators in more intuitive units (e.g., hand sizes, tablespoons) for each MFQ category to reduce participant burden and make the questionnaire easier to complete. For this, we used guidelines from the British Nutrition Foundation and the British Dietetic Association [14,15]. Serving sizes that could be counted in units were presented as such (e.g., 3 fish fingers).

Two researchers split the work on the different animal groups between them and worked independently throughout the whole development process, meeting regularly to discuss and resolve queries, and cross-check all work.

### 2.3. MFQ Format

The MFQ is designed to be used in an online setting (Appendix A). Individuals completing the MFQ are first presented with instructions on how to fill in the questionnaire (Appendix A). They are asked to select which of the following animal types, if any, they consumed on the previous day: beef, pork, chicken and turkey, lamb and mutton, game, and fish and seafood. Individuals are then asked to indicate how many servings they consumed of each MFQ category on the previous day, on a scale from 0 to 9, allowing for half servings. An ‘Other’ category for each animal type allows individuals to add further items not covered in the questionnaire.

### 2.4. Reliability Study

To assess the test–retest reliability of the questionnaire, we conducted an online 3-week longitudinal study—hosted on the survey website Qualtrics (https://www.qualtrics.com; accessed on 2 November 2020). The study covered two data collection weeks with one washout week in between. We chose a washout period of one week to reduce the risk of participants changing their diets between assessments. During the data collection weeks, participants were asked to complete the MFQ daily. The primary outcome of the study was the test–retest reliability of the MFQ with regard to the mean daily meat intake from all sources. The study protocol was granted ethical approval by the Medical Sciences Interdivisional Research Ethics Committee (IDREC) of the University of Oxford (REF: R71783/RE001) and the study was conducted in accordance with the Declaration of Helsinki.

### 2.5. Participants

For the sample size estimation, we were guided by the dietary assessment literature. In their methodology review, Cade and colleagues recommend a sample size of at least 100 participants to compare agreement between methods using the Bland-Altman method [3]. Assuming a 30% dropout rate, we recruited 140 participants to our study. As we did not conduct a power calculation specifically for the primary outcome, the intraclass correlation coefficients (ICC) for total meat intake, the 95% confidence intervals reported here should be used to interpret the realized power of our study.

Participants were recruited to the study through the online research recruitment platform Prolific Academic (https://www.prolific.co; accessed on 1 November 2020) and screened for eligibility before starting the study. To be eligible, participants had to be aged ≥18 years, be willing and able to give informed consent for participation in the study, self-report to eat meat at least five times per week (regardless of portion size), not actively be trying to reduce their meat consumption, not be enrolled in any other dietary intervention study, not be trying to lose weight, self-report to speak English fluently, and be a resident in the UK. Eligible participants were invited to take part in the full study and were asked to read the participant information sheet, informing them that they were free to withdraw at any time, and indicate their consent to take part. During data collection weeks, participants were sent daily automated private messages to their Prolific accounts, asking them to complete the MFQ, each time reflecting back on the previous day. To minimize attrition bias, questionnaires were kept open for 72 h to allow participants to complete them up to two days later. At the start of the study, participants completed a baseline questionnaire asking about their demographic characteristics and perceived meat identity. A brief questionnaire at the end of the final session asked participants to evaluate the MFQ. Participants were reimbursed GBP 2.50 for completing the first questionnaire, and GBP 1 for each of the following questionnaires, totaling up to GBP 15.50 by the end of the study.

### 2.6. Statistical Analysis

The analyses for the reliability study were conducted in R (version 3.4.1, R Development Core Team 2017, University of Auckland, Auckland, New Zealand), Excel 2016 (Microsoft Corporation, Washington, DC, USA), and Stata/IC (version 14.1, StataCorp LLC, College Station, TX, USA). A statistical analysis plan was published on the Open Science Framework preceding the analyses [16].

### 2.7. Data Preparation

Participants who completed <4 questionnaires in either Week 1 or 3 were excluded from the analysis, as meat consumption varies significantly day-to-day and aggregated data of less than four days might not be reflective of actual weekly consumption. We assessed the ‘Other’ entries for each animal category and assigned them to existing categories where appropriate (e.g., adding “King Prawn (curry) 250 g” to Fish and Seafood Moderate Mixed Dish). There were two products which were not yet captured by the questionnaire: lamb burgers and game burgers. We updated the questionnaire with these two items and calculated their mean meat content per serving using the burger values of the chicken, pork and beef categories. Where participants entered a serving quantity for the ‘Other’ items without describing the product or its serving size, we used the mean meat content per serving of the respective animal category for our analyses.

Mean total daily meat intake was calculated for each participant and each data collection week, using the underlying meat proportions of the MFQ. This was also calculated separately for each animal category and the subgroups red meat, processed meat and, in combination, red and processed meat. Red meat comprised all beef, pork and lamb items of the questionnaire. Processed meat included all items that cover coated products, slices, bacon, pies and pastries, sausages, sausage rolls, and burgers of all animal categories, apart from fish and seafood. We also calculated the total mean daily consumption for weekday versus weekend data for each week.

### 2.8. Sample Exploration

Demographic characteristics of the sample and adherence to the daily completion of the MFQ was explored descriptively.

### 2.9. Primary Analysis

The test–retest reliability of the mean total daily meat intake was assessed through ICC, using absolute-agreement and single rater two-way random models [17]. This type of ICC analysis allowed us to generalize the findings to a wider population of ‘raters’, i.e., UK residents with similar demographic characteristics to our sample. ICC values were interpreted as follows: ≤0.20 = poor agreement, 0.21–0.40 = fair agreement, 0.41–0.60 = moderate agreement, 0.61–0.80 = good agreement, and >0.81 = excellent agreement [18,19]. We created Bland-Altman plots to visualize the test–retest reliability, based on the mean difference and 95% limits of agreement (LOA) of the two measures [20]. The Bland-Altman plots were created using log-transformed difference data because the differences between weeks were not normally distributed and a strong magnitude bias existed between means and differences of the two weeks [21]. Linear regressions on the log-transformed difference data were employed to assess the relationship between the bias and magnitude of the measurement [21].

### 2.10. Secondary Analysis

The primary analysis was repeated using the mean daily meat intakes of the subgroups red meat, processed meat, red and processed meat, chicken and turkey, and fish and seafood. To assess for weekday/weekend differences in test–retest reliability, we repeated the primary analysis using mean total daily meat intake calculated from weekday vs. weekend data, respectively. To assess the effect of outliers, we ran a sensitivity analysis excluding participant’s data from days on which their reported meat consumption was >1.5 kg of meat.

The evaluation questionnaire was assessed descriptively. Written feedback provided voluntarily by participants as part of the evaluation questionnaire was analyzed qualitatively using inductive thematic analysis [22]. Following familiarization with the data, themes were generated from recurring topics. No a priori framework or ideas were imposed onto the data and we only recorded themes that emerged from participants’ comments.

## 3. Results

### 3.1. Sample

We recruited 140 participants to the study, 133 participants completed questionnaires in both Week 1 and Week 3. Four participants were excluded because they had less than four entries for either Week 1 or Week 3; therefore, the final sample included 129 participants. The sample represented the UK population well in terms of age, gender, ethnicity, region of residence and household size (Table 1). Adherence was very high in both weeks (90% and 88% of participants completed all seven sessions in weeks 1 and 3, respectively). For the outlier sensitivity analysis, we excluded daily meat intake data >1.5 kg; this affected four data points across the study. The participants with outlier data all had at least four other completed questionnaires in the week of the outlier and could therefore still be included in the analysis. Across all measures, apart from beef and processed meat, participants reported higher consumption in Week 1 than in Week 3 (Table 2).

### 3.2. ICC and Test–Retest Reliability

The ICC analysis revealed good reliability of the questionnaire for the primary outcome: mean total daily meat intake (ICC = 0.716, CI 0.621, 0.788). The subgroup analyses showed good reliability for the processed meat, red and processed meat, and poultry outcomes, and moderate reliability for the red meat outcome. The ICC for fish and seafood indicated fair reliability (ICC = 0.257, CI 0.118, 0.387; Table 3). The mean absolute difference (bias) in mean daily meat intake between Week 1 and Week 3 of the MFQ was 12% (Figure 2, Table 4). After log transforming the difference data, regression analyses for nearly all outcomes showed no association between level of meat intake and differences between weeks 1 and 3. Exceptions were found for poultry, fish and seafood, and weekend data, for which higher levels of intake were associated with a greater decrease in meat consumption from Week 1 to Week 3, (Table 4).

### 3.3. Instrument Evaluation

The majority of participants rated the instrument as very easy to complete (Figure 3). We explored Qualtrics data on the time it took participants to complete the questionnaires for Week 1 Days 2–7 and Week 3 Days 1–6 (i.e., sessions without additional tasks). Descriptive analysis of this completion time data (*n* = 1503) revealed that participants took on average (median) 71 s (IQR = 66) to answer the MFQ, with virtually no difference between the weeks (Week 1: Median = 68 s, IQR = 64 s; Week 3: Median = 74 s, IQR = 68 s).

We asked participants whether anything had been left unclear in the questionnaire. Of the 99 participants who answered this optional question, 94 said that everything had been clear. Participants were able to provide us with more feedback on the questionnaire in a comments text box. Thirteen said that completing the questionnaire daily for two weeks made them reflect on their meat consumption. For some, the self-monitoring data was surprising.


*“It actually just made me realise how often I eat meat, every single day apparently!”*



*“This has kind of made me realize I eat more meat each week than I thought.”*


For three participants, this reflection prompted decisions to take action, including wanting to reduce meat intake or wanting to continue keeping a food diary.


*“It’s made me think about the amount of meat I eat, and following a discussion with my wife, we have now decided to start with 1 no meat day a week and work towards more.”*



*“The study has made me think about what I am eating and I have promised myself to keep a food diary going forwards.”*


Next to stating that the questionnaire had been clear, 12 participants also explicitly commented on the fact that they had found it easy to do. Eight participants found it enjoyable to track their meat consumption.

Four participants mentioned that completing the questionnaire took some getting used to, but all four concluded that it had been easy after a bit of practice.

A few participants raised issues regarding the questionnaire content and format. Three participants stated that they found the instructions quite complex and they, therefore, had to be careful to complete the questionnaire correctly.

Three participants stated that they found the meat categories unclear or had consumed meat products that they felt were not reflected in the questionnaire. Two felt it was difficult to estimate the correct serving size.


*“Some food stuffs I felt were unclear—does dried sausage such as saucisson sec or kabanos count as a ‘slice’, or a sausage, or in the sausage roll category?”*


Some participants commented on issues with the questionnaire format. Three found that they sometimes accidentally indicated having had servings of items they did not consume, especially if they completed the questionnaire on their phone. Two others found that the presentation of the items in separate scrollable boxes for each animal category, and all on the same page, was confusing.


*“When scrolling on my mobile, sometimes I would catch the sliders for other answers. It was a minor annoyance but could lead to an incorrect entry if a participant didn’t notice they’d done it before submitting their answer.”*



*“Could easily put a poultry sausage in when it was meant to be a pork sausage, for example, as you see “sausage” (which in the UK is usually pork), forgetting you are in the poultry section and the pork section is below.”*


## 4. Discussion

The MFQ offers a new and quick approach to measure meat intake at a granular level. It was developed using UK data on standard food portion sizes and meat content of composite dishes and is therefore tailored to a UK population. Our test–retest reliability study revealed moderate to good agreement on all key meat intake measures, and fair agreement for fish and seafood. The MFQ may therefore be appropriate to use in intervention studies, which aim to measure changes in meat intake. The evaluation showed that most participants found the questionnaire easy to use and managed to complete it in less than 2 min.

It is difficult to compare our results to those of other FFQs due to the MFQ’s unique focus and novel testing regimen. To the best of our knowledge, the other two meat FFQs that have been developed so far have only been assessed for relative validity, and not for reproducibility [9,10]. We could not find questionnaires that, like our MFQ, measure weekly consumption through several 24 h questionnaires. In comparison to other self-administered and partly web-based FFQs, which were completed twice within a 1–6 week interval and measured food consumption for the preceding month, our mean ICC of 0.598 is similar (mean ICCs of 0.63 [23], 0.72 [24], and 0.79 [25], respectively). Though in contrast to our study, these studies assessed reproducibility by comparing intakes of nutrients, and in one case fruit and vegetable servings [25], not intake of meat.

The one measure with only fair agreement in our MFQ belonged to the fish and seafood category. This finding is in line with another FFQ reliability study, which also found considerably lower ICC results for its seafood category [26]. The low agreement is likely to be due to more sporadic consumption of fish and seafood compared to other types of meat, as shown in the UK-wide NDNS survey [27]. Likely, one week of 24 h measurements did not accurately reflect usual consumption and hence differences were found between the data collection weeks. Given the lower ICC, the reliability of the MFQ for the measurement of fish and seafood intake, specifically, is limited.

Our sensitivity analysis revealed that weekday data was more reliable than weekend data, as would be expected given the different number of days involved. This could be due to the fact that weekends tend to have more variable dietary patterns than weekdays [28,29], as evidenced by the larger standard deviations. We therefore recommend using average weekly data as the main outcome to allow for differential patterns of consumption across a week.

Overall, we measured a reduction in total meat consumption from the first to the third week of the study. We can only speculate on the reasons for this. It is conceivable that this is due to flagging motivation, leading to less thorough answers. However, we found that the completion time for Week 1 and Week 3 questionnaires was virtually the same, so it seems unlikely that our participants were less thorough in the second round of testing. Instead, it is possible that the reduction was due to a self-regulation process [30,31], meaning that self-monitoring meat intake may have led participants to reflect on their meat consumption, compare it to their meat consumption goals, and align their behaviour with their ideal (reduced-meat) state. Indeed, previous research has found that a simple combination of self-monitoring red meat intake with goal setting can lead to significant reductions in meat consumption [32,33]. In our qualitative analysis, we found that participants commented on the fact that self-monitoring their meat intake made them realize and think about their meat consumption, and three participants indicated they had decided to take action. This further supports the idea that completing the MFQ might have had a mild intervention effect.

Another explanation for the higher measures in Week 1 and lower standard deviation in average consumption of Week 3 could be that participants made more accidental entries at the beginning of the study. Indeed, the four outliers we removed in our sensitivity analysis (>1.5 kg daily consumption) were all measured in the first week. Participants mentioned in their feedback that the presentation of the questionnaire occasionally led to accidental entries. Indeed, the outliers we identified included unlikely high servings of one item (e.g., 9 servings of beef cuts). Participants fed back to us that it took them a while to get used to the MFQ, but that it got easier over time. In future studies, it might be worthwhile introducing participants to the MFQ for a few days before starting actual data collection, to minimize unnecessary errors. Moreover, another option could be to add a confirmation screen asking participants to confirm their indicated consumption. When removing the four outliers from our data, the ICC for mean total daily meat intake increased from 0.716 to 0.777.

## 5. Strengths and Limitations

To the best of our knowledge, the MFQ is the first meat frequency questionnaire to capture individuals’ meat intake in a comprehensive and detailed fashion. Our participants found it easy and quick to complete, and some even found the task enjoyable.

One limitation of the questionnaire itself is that the data underlying the standard portion sizes of the questionnaire is taken from the FSA food portion sizes book, which was last updated more than a decade ago [12]. The book likely underestimates true portion sizes nowadays, as portion sizes have steadily increased over the years and consumers tend to eat more than recommended serving sizes on packs [34,35]. To alleviate this issue, we provided participants with simple visual cues of the serving size of each meat item (e.g., 3 fish fingers or steak cuts the size of your hand) and allowed them to indicate half servings. We thus aimed to enable participants to make accurate entries despite potentially smaller than average serving sizes. As the MFQ focusses on meat items found in FSA and NDNS data sets, it represents a typical UK diet, and might not be appropriate for usage in other countries. However, we provide the MFQ together with its underlying meat proportions in Appendix A, which researchers can adapt to use in other contexts/countries. Another limitation is that the MFQ requires daily assessment across a whole week, which makes it more burdensome than one 1-week FFQ. However, our MFQ is still less burdensome than food diaries, and we found that our participants needed only 71 s on average (median, IQR = 66) to complete the questionnaire each day.

Our test–retest reliability results indicate that the questionnaire may be appropriate for use in studies aiming to assess changes across time. The sample recruited to the test–retest reliability study was representative of the UK adult population on the outcomes we measured. Our findings should therefore be generalizable to other UK adult samples, though we cannot comment on the representativeness of our sample in terms of socioeconomic status. Since we did not assess relative validity, we cannot make statements about the accuracy of the absolute values of meat intake. Future research may wish to compare the MFQ with other gold-standard measures of dietary intake to assess this. Moreover, we studied people who consumed meat at least five times per week. Future studies may wish to assess the reliability of this questionnaire in a population of low-meat eaters. In addition, we have not tested this instrument in an adolescent population which may be an interesting topic for further research. Other limitations include the short washout period between data collection weeks, due to which we cannot rule out any memory effects. Moreover, by keeping the questionnaire open for 72 h, recall bias was potentially increased.

## 6. Conclusions

The new MFQ offers a quick, acceptable and reliable method to measure changes in individuals’ meat intake in a UK population. Key meat intake outcomes, such as red and processed meat consumption, are reliably captured. Only the fish and seafood category had lower levels of agreement. We hope the questionnaire will be useful to other researchers in the environmental and health behaviour change field, who wish to capture changes in meat consumption.

## Figures and Tables

**Figure 1 nutrients-13-00922-f001:**
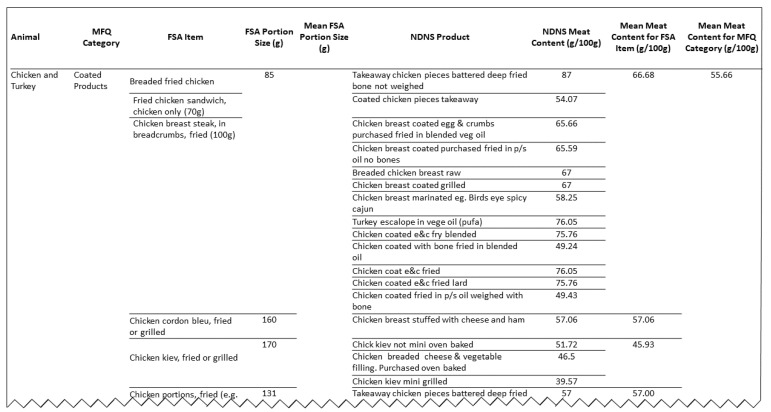
Extract of the underlying data for the chicken and turkey category of the Oxford Meat Frequency Questionnaire (MFQ). Extract depicts the underlying portion size information from the Food Standards Agency (FSA) food portion sizes book and the underlying meat content information (g/100 g serving) from the National Diet and Nutrition Survey (NDNS) Year 10 food composition database.

**Figure 2 nutrients-13-00922-f002:**
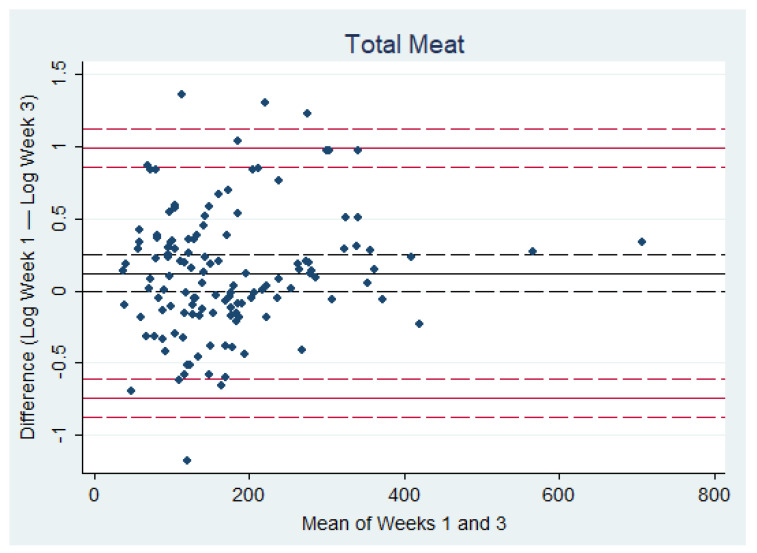
Bland-Altman plot for total meat (g/day) showing the comparability of weeks 1 and 3 of the Oxford Meat Frequency Questionnaire. Measurements of meat intake after log transformation of difference data plotted against original mean data. Limits of Agreement are shown as solid red lines with 95% confidence intervals (dashed lines). The mean difference (bias) is shown as the solid black line with 95% confidence intervals (dashed lines). The mean difference (bias) data is on a log scale and so is equivalent to a ratio on a linear scale.

**Figure 3 nutrients-13-00922-f003:**
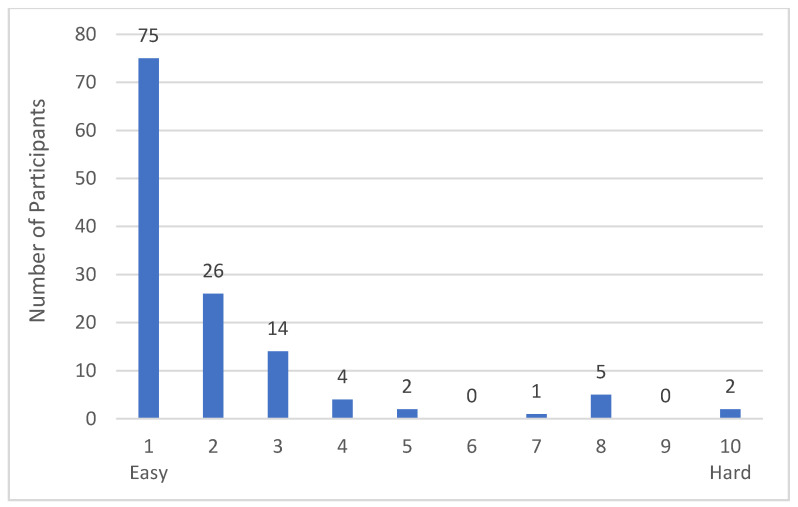
Participants’ difficulty rating of the Oxford Meat Frequency Questionnaire. Responses from study participants to the evaluation question “How easy or hard did you find filling in the questionnaire? (Scale from 1 easy to 10 hard)”, collected on the last day of the study (Day 7, Week 3). *n* = 129.

**Table 1 nutrients-13-00922-t001:** Characteristics of study participants (*n* = 129).

	Study Participants
Age, median (IQR) (Minimum–Maximum: 18–76 years)	37 (23)
Gender, *n* (%)	
Female	60 (48.0)
Ethnicity, *n* (%)	
White-British	103 (79.8)
White Other	14 (10.9)
Asian or Asian-British	5 (3.9)
Black or Black-British	4 (3.1)
Mixed/Other	3 (2.3)
UK Region of Residence, *n* (%)	
South East	26 (20.2)
East Midlands	15 (11.6)
North West	14 (10.9)
Greater London	12 (9.3)
Yorkshire and the Humber	11 (8.5)
Scotland	11 (8.5)
West Midlands	10 (7.8)
South West	9 (7.0)
East of England	7 (5.4)
Wales	6 (4.7)
North East	5 (3.9)
Northern Ireland	3 (2.3)
Household Size, *n* (%)	
1	28 (21.7)
2–3	66 (51.2)
4–5	30 (23.3)
6+	5 (3.9)
Meat Identity, *n* (%)	
Omnivore	81 (62.8)
Meat Eater	47 (36.4)
White Meat Only	1 (0.8)
Pescatarian	0 (0.0)
Flexitarian	0 (0.0)
Dairy-free	0 (0.0)
Vegetarian	0 (0.0)
Plant-based	0 (0.0)
Vegan	0 (0.0)
Meat Identity Since, *n* (%)	
For more than 2 years	129 (100.0)
For 1–2 years	0 (0.0)
For 6–12 months	0 (0.0)
For 1–6 months	0 (0.0)
For less than a month	0 (0.0)

Data are median (IQR) or *n* (%).

**Table 2 nutrients-13-00922-t002:** Mean daily intake of meat categories in Week 1 and Week 3 as assessed by the MFQ.

	Week 1		Week 3	
	Mean (SD)	% of Total Meat	Mean (SD)	% of Total Meat
Total Meat	189.0 (126.7)	-	161.9 (94.5)	-
Red Meat	84.8 (57.9)	44.9%	78.2 (55.5)	48.3%
Processed Meat	42.0 (36.6)	22.2%	43.0 (44.1)	26.6%
Red and Processed Meat	126.8 (81.4)	67.1%	121.2 (88.4)	74.9%
Poultry	76.7 (80.4)	40.6%	64.7 (60.9)	39.9%
Fish and Seafood	24.5 (47.6)	13.0%	17.9 (23.4)	11.0%
Beef	33.6 (39.5)	17.8%	33.6 (34.2)	20.8%
Pork	44.4 (41.6)	23.5%	38.6 (36.6)	23.8%
Lamb	6.8 (16.0)	3.6%	6.0 (17.4)	3.7%
Game	2.9 (18.3)	1.5%	1.2 (6.3)	0.7%
Weekday	191.0 (134.2)	-	162.7 (101.4)	-
Weekend	183.4 (157.1)	-	163.3 (110.3)	-
Total Meat Without Outliers	182.1 (119.0)	-	161.9 (94.5)	-

Data are mean (SD) intakes in g/day and % contributions of meat sub-types to total meat (*n* = 129). The outlier sensitivity analysis excluded daily meat intake data >1.5 kg. MFQ: Oxford Meat Frequency Questionnaire.

**Table 3 nutrients-13-00922-t003:** Intraclass correlation coefficients (ICC) between daily mean intakes of meat assessed by Weeks 1 and 3 of the MFQ.

	Measure	ICC	ICC Interpretation	*p*-Value	Confidence Interval ICC
**Primary Outcome**	Mean Total Daily Meat Intake	0.716	good	<0.001	0.621, 0.788
**Secondary Outcomes**	Red Meat	0.531	moderate	<0.001	0.419, 0.628
	Processed Meat	0.650	good	<0.001	0.558, 0.727
	Red and Processed Meat	0.677	good	<0.001	0.591, 0.749
	Poultry	0.680	good	<0.001	0.592, 0.752
	Fish and Seafood	0.257	fair	0.001	0.118, 0.387
**Sensitivity Analyses**	Without Outlier Data	0.777	good	<0.001	0.703, 0.832
	Weekday	0.649	good	<0.001	0.547, 0.730
	Weekend	0.448	moderate	<0.001	0.325, 0.556
**Mean**		0.598			

Absolute-agreement and single rater two-way random models were used to assess intraclass correlation coefficients (ICC). ICC values were interpreted as follows: ≤0.20 = poor agreement, 0.21–0.40 = fair agreement, 0.41–0.60 = moderate agreement, 0.61–0.80 = good agreement, and >0.81 = excellent agreement [18,19]. MFQ: Oxford Meat Frequency Questionnaire. *n* = 129.

**Table 4 nutrients-13-00922-t004:** Linear regression analyses for magnitude bias for quantifying meat intake in Weeks 1 and 3 of the MFQ.

	Mean Difference (Bias)	Regression Coefficient	*p*-Value	95% CI
Mean total daily meat intake	0.12	0.08	0.271	−0.06, 0.21
Red meat	0.09	−0.05	0.615	−0.26, 0.16
Processed meat	0.01	−0.00	0.969	−0.22, 0.21
Red and processed meat	0.06	−0.01	0.909	−0.17, 0.15
Poultry	0.05	0.25	0.015	0.05, 0.45
Fish and Seafood	0.15	0.47	0.038	0.03, 0.91
Sensitivity analysis–weekday	0.13	0.05	0.503	−0.10, 0.21
Sensitivity analysis–weekend	0.02	0.20	0.032	0.02, 0.38
Sensitivity analysis–without outlier data	0.10	0.03	0.620	−0.10, 0.16

The mean difference (bias) is on a log scale and so is equivalent to a ratio on a linear scale. The values here therefore represent the percentage difference between weeks 1 and week 3. MFQ: Oxford Meat Frequency Questionnaire; CI: Confidence Interval. *n* = 129.

## Data Availability

The data presented in this study are available from the corresponding author on reasonable request.

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
