# Peer review of "Development and Reliability of the Oxford Meat Frequency Questionnaire"

_nutrients, 2021, doi:10.3390/nu13030922_

Round 1

Reviewer 1 Report

This very interesting manuscript describes a new instrument for measuring meat consumption, something that I think is very much needed, setting out its development, reliability assessment, and feedback from participants. The work here establishes an instrument with reasonable properties, creating a platform for (what I would have assumed to be the next stage of) higher forms of validation. An analysis plan was published on 4 December 2020, which is commendable and suggests an admirably efficient analysis process.

Overall, I like this manuscript a lot and I think it could make an important contribution to the literature. There are a few concerns that I have, mostly of a statistical nature, but I don’t imagine that these will present too much difficulty to the authors in addressing. I will first describe my more substantial queries and then list some minor/specific points.

The manuscript is well-written and sets out a clear motivation for the development of the FFQ here. Taking a step back, though, I think an initial paragraph in the introduction that motivates why we want to measure food intake in the first place (associations with disease [which is explained in the context of meat consumption on Lines 35–37, so I’m suggesting a more general statement here], assessing impact of interventions [which is well covered later on under materials and methods, so I’m just suggesting a mention here], etc.) followed by a paragraph that explains the alternatives (e.g., FFQs, weighed diaries, recalls) and their strengths and weaknesses would be helpful to the reader in establishing the context of your research. Then the currently first paragraph hones in on your specific topic. At the moment, the manuscript seems, to me, to skip much of the motivation behind your study, which will be familiar to many, but I suspect not all, of your potential readers, and even those who are familiar with the field might still benefit from a quick recap/reminder. Aside from this, I thought that the introduction was excellent in efficiently motivating the specific need for this instrument.

I felt that the one week washout (Lines 16 and 126–127) was perhaps a little short to avoid memory effects, and I was expecting that would be probably be included as a limitation in the Discussion but didn’t see it mentioned there, but at the same time, I appreciate that this reduces the risk of the participant changing their diet between assessments and should have contributed to retention of participants. On the basis that I can see other readers wondering about these trade-offs, I wonder if a little more information could be given around Line 126 about this decision, with any resulting limitations added to the Discussion?

While I appreciated the upfront acknowledgement on Line 134, there are “more statistical” ways to justify the sample size for the Bland-Altman method (see DOI: 10.1016/S0140-6736(86)90837-8 and https://www-users.york.ac.uk/~mb55/meas/sizemeth.htm, but note that this approach doesn’t directly address type 2 errors, whereas DOI: 10.1515/ijb-2015-0039 does) than Lines 135–136 and Cade, et al. is a little more vague in their suggestion (effectively: at least 50, preferably much more, say 100) than your statement on Lines 135–136 suggests to me. The median size of 110 in the literature isn’t for me an argument either way about the correct sample size. Given that the primary outcome, however, was the ICC for total meat, and concerns reliability, I would have anticipated that the study would be designed to address that particular question (e.g. DOI: 10.1002/sim.1108). Based on this approach, for n=112 (80% of the 14 on Line 138), the 95% CI for an ICC of 0.705 (in the middle of “good”) would be 0.61–0.80, so spanning the full range of this category, with anything a little higher or lower becoming at least slightly ambiguous in its interpretation as moderate/good or good/excellent. Needless to say, this can’t be altered now, but perhaps a sentence along the following lines might help to anticipate such questions from readers: “The power calculation did not directly address the primary outcome, the ICC for total meat intake, and the 95% confidence intervals here should be used to interpret the realised power of our study.”

I wondered a little about the requirement that participants must eat meat at least five times per week. I would expect this to bias the ICC estimates downwards (via the reduced between-participant variability and given the lack of rarity of such levels of consumption) but it also raises questions for me about the external validity of the results. If the goal is to develop an instrument for interventions that might reduce meat intake, this would be a reasonable criterion here, as the inclusion criteria for such a study would be likely to require regular meat consumption, but this seems to weaken the generalisability for more broad use. This seems worth justifying here (Line 142) and revisiting in the Discussion.

I was also less convinced about excluding those with fewer than 4 questionnaires (Lines 159–161) which for an effectiveness trial, which meat-reducing interventions will almost certainly be, would still require their data to be analysed. It’s slightly unclear to me whether the “respectively” on Lines 159–160 means at least 4 questionnaires in each week or in total, but I’m guessing the former. This seems likely to be a feature of pragmatic use of the instrument, and excluding those with fewer days’ of data would seem to me likely to bias the ICCs upwards. It’s not clear to me how their data “might be biased” (Line 161, more variable yes, but the mechanism for bias alluded to isn’t clear to me) and perhaps some more explanation on this point could be added around here. In any case, I think repeating the analyses using all available data would be helpful for readers. Even if this reflects a “worst case” in terms of compliance, it is a situation that researchers using your instrument will have to address at some stage. I also feel that the approach currently taken here is inconsistent with the weekday versus weekend analyses (Lines 195–196, the number of days required was not stated there but with only 2 weekend days per week, this seems likely to be lower, in terms of the mean number of days, than the requirement of 4 as described above even if that was for the two weeks combined)

The manuscript describes using the two-way random model (Lines 184–185), which would not be my recommendation (this is speaking as a biostatistician who is asked this very question on occasion) for a test-retest study. The two-way mixed effects model is more conventional, see e.g. the well-cited article DOI: 10.1016/j.jcm.2016.02.012, as the measurement occasions are better seen as fixed and not random, as raters might well be (and participants here are). Can the authors explain/justify their choice, or perhaps change their description to the two-way mixed effects model instead? Note that the importance of this decision is in terms of the interpretation rather than the calculation of the ICC.

Note that ICCs should perhaps be accompanied by confidence intervals in the text as well as in the table. The p-values in Table 3 and on Line 222 and 225 are not particularly helpful in my view, since they are testing the null hypothesis of an ICC of zero (which at the population level would mean that there was no between-subject variance, the sample estimate can of course be estimated to be zero, or even negative, irrespective of the population ICC) and this is highly implausible from the outset. The fact that the p-value is, but for the inequality sign, the same for fish and seafood as for total meat further shows, for me, the lack of information in these values.

Given the clear, to me, patterns (trends and heteroscedasticity) in the Bland-Altman plots (Figure 2 and supplementary panel Figures S5 and S6, I’m not sure that the LoAs are, in fact, valid/useful here. The values on Line 229 (29g, -128 to 186g) would only apply to someone with an intake around 200g (based on visual inspection) and not to someone with an intake around 50g or 400 g. Personally, I wouldn’t report these as an overall measure of the differences between the two assessments given this heterogeneity. It is possible that log transformations could correct both the variance and maybe, at least some of, the bias issues. You could look at DOI: 10.1191/096228099673819272 and DOI: 10.1002/sim.5955 for some suggestions of what could possibly be done here, or most simply show the graphs without the LoAs and allow the reader to interpret the implications of the disagreement between the two assessments not being constant over intakes. I think it would be preferable to take one of the more sophisticated options though and so in anticipation, note that there is considerable value in providing CIs around the bias (e.g. Table 4 and Line 229) and the LoAs (see DOI: 10.1191/096228099673819272 again), even if these are only shown in figures and/or tables and not referred to in the text.

As a final suggestion, I wonder if readers might not also be interested in the ICC for a single day’s measurements from the first week’s assessments (reducing the risk of the instrument being an accidental intervention, as you note on Lines 344–353). While you appropriately note the issue of the weekday versus weekend effects (recommending using both on Lines 337–338), you could report ICCs overall and separately for these. This would allow a reader to get some idea of the expected reliability, ceteris paribus, for, say, three weekday and one weekend, which would allow some flexibility around participant burden. You raise the challenges around a week’s worth of data on Lines 381–382.

Some specific, often minor comments are below:

Line 9: While I agree, I think “Reliable and valid measurements…” would be a more complete statement of this. I appreciate that this isn’t among the goals here, but it is true nonetheless and without validity, reliability has no value.

Lines 16–17: I’d specify the ICC form being used here.

Line 18: Rather than the mean ICC, which I think is difficult to interpret, particularly here as it includes sensitivity analyses, perhaps you could give the minimum–maximum (0.53–0.68) for the secondary outcomes excluding fish and seafood, which you then describe? Otherwise, I think “average ICC” here is likely to cause confusion for some (most?) readers without further explanation of what this means. I suggest always specifying what average is intended (mean, median, etc.), here in the abstract and throughout the manuscript. Note that you sometimes use “mean” already, e.g. Lines 179, 188, 228, ; Tables 1, 2, 3, 4; Figure 2; supplementary panel Figures 5 and 6; and perhaps elsewhere.

Line 22: Purely as a suggestion, I wonder if it would be useful to indicate the next step here, e.g. “The next step is to validate the instrument against…”?

Line 51: Given the age requirement, I’d suggest “in a UK sample of adults.” here. The use of the instrument in adolescents would be an interesting topic for further research, I think, given their mixture of household-dependent and independent food choices and you could raise this in the Discussion if you agreed.

Line 58: I wonder if adding “although such an instrument could also be used in observational research” would be useful here to encourage readers to make full use of the instrument (I appreciate that “for instance” here indicates that this is not an exhaustive list). I can’t see any flaws in your instrument that would be specific to particular study designs.

Lines 58–59: If you wanted to give an example, you could mention semi-vegetarian diets and perhaps more concretely “meat-free Mondays” as a campaign that would be expected to increase day-to-day variability here, and/or the traditional weekday versus weekend dietary patterns.

Line 60: “avoid” is a very strong claim, needing only a single counterexample to disprove, so perhaps “minimise” or similar instead? See also Line 147 and perhaps elsewhere.

Line 93: “average” (see comment above). See also at least some of Lines 95, 98, 99, 129, 166, 168, 170, 183, 192, 195, 222, 228, 256, 321 (twice), 336, 337, 355, 364, and 384; Tables 2, 3, 4; and perhaps elsewhere.

Lines 129–132: I’m assuming that participants indicated their consent either prior to or by completing the first assessment (Lines 421–422 isn’t clear on the timing) and were free to withdraw at any time, but I think this might be worth a sentence around here.

Lines 147–148: While true, this would also seem to trade-off expected recall bias for this reduction in expected attrition bias.

Line 152: Wouldn’t this be £15.5, which given your use of pence on Line 151 doesn’t seem excessively precise?

Lines 197–198: This might be clearer, assuming I’m understanding what was done, as “excluding … > 1.5kg” (c.f. Lines 209–210).

Line 199: Just a suggestion, but I’d delete “intervention” here, and also Lines 251, 261, and 312, and replace “intervention” with “instrument” on Line 252. I know that you’ve designed the instrument for an intervention context, but I couldn’t see how completing it counted as an intervention in itself (your comment on Lines 352–353 suggests an accidental effect rather than anything else, and which is a risk for all questionnaires, interviews, etc.).

Line 209: Since the IQR doesn’t communicate much here (beyond at least 75% having 7 days’ data), perhaps the percentage with all 7 days would be more useful?

Line 226: Perhaps “differences” (as you use on Line 228) rather than “changes” (which sounds more causal/definite to me) as you don’t know that the true intakes changed, changes in reporting could also produce this effect. Also Line 19 back in the abstract.

Line 255: It might improve the argument here if you indicate the largest remaining value (if it was 300 seconds, for example, this would indicate a clear separation). If you present medians and IQRs, as suggested below, the outliers should become effectively irrelevant.

Line 256: I’m assuming this “average” is a mean (rather than one of the other forms of average). This should be made explicit with SDs added. However, as indicated on Lines 257–258, this data is, as might be expected, very skewed, and so medians and IQRs might be better on Lines 256–258 and 384. Note that the variability in the times presented here is a mixture of between- and within-participant variability and not easy to interpret if all n=1493 data rows were included. I’d be inclined to calculate a per day mean per participant and present the median and IQR (or mean and SD if this was close enough to a normal distribution), assuming that the values here don’t already represent a summary measure of course.

Line 282: I’m not sure that this quote adds anything above and beyond Lines 280–281. The same also applied, to me, for Lines 285–286 which seemed to me to effectively repeat Lines 283–284.

Line 316: I’m always cautious about absolute statements (both there being only two and the extent of their evaluation here) and would suggest adding “To our knowledge” at the start of this sentence. (You are, what I would regard as, appropriately cautious on Lines 317–319 and 366.)

Lines 333–224: I think this is more predictable, simply given the use of a mean of up to 2 observations versus a mean of up to 5 observations (the same would apply for the mean of 2 measurements of waist circumference versus 5, for example, or measurements of systolic BP, etc.), than all readers might immediately appreciate here. Perhaps you could say: “Our sensitivity analysis revealed that weekday data was more reliable than weekend data, as would be expected given the different number of days involved. This could also be due…” You could then, if you agree and so wished, delete Lines 335–337.

Lines 361–363: Again, I’d replace “avoid” with some less absolute word (“minimise”, “reduce”, etc.). I also wondered if a confirmation screen summarising their non-zero entries that they have to submit or go back and edit these entries might be another strategy here.

Lines 385–386: You haven’t looked at the validity/responsiveness of the instrument, so I think the wording here goes a little too far. There are some additional steps that I would want to see achieved before I’d use the instrument myself. You move on to some of these on Lines 388–391, so perhaps this is more a request for some “softening” of the first sentence of the paragraph and, if you agree, a strengthening of the need for studies to look at construct validity. I think you could argue that face and content validity are covered already if you wished.

Line 395: I would read “appropriately captured” as claiming validity and not just reliability.

Table 1: Range is strictly speaking the difference between the minimum and maximum (https://en.wikipedia.org/wiki/Range_(statistics)) so I would describe these values as “Minimum–Maximum” instead). It seems that age is positively skewed and/or contains outliers (the lower limit for a 95% reference range would be 11.2, well below the minimum age of 18), so perhaps median (with an IQR) might be more informative about the typical age of participants? Or perhaps age categories might be more/also informative? I’m not sure that the bottom six categories for “Meat identity” need to be listed, or at least listed separately, given they are all zero. The same applies to the time since this identity where there are four rows of zero. I also wondered if the household sizes are a little detailed, e.g. would 1, 2–3, 4–5, and 6+ still work well enough?

Table 2: Again, several of these are clearly skewed to the right and/or have outliers (whenever the SD is more than half the mean, the lower 95% reference range limit will be negative). Perhaps medians (and IQRs) would be more useful? I found myself calculating the relative contributions of meat types to total daily intake and wondered if that compositional information could be added to the table.

Table 4: Try to avoid the asterisk system for statistical significance and instead give p-values (my preference) or omit and interpret the 95% CIs instead.

Figure 2 and supplementary panel figures 5 and 6: The direction for the y-axes should be made explicit. Are these week 1 values minus week 3 values? If so, the other way around might be clearer.

Reviewer 2 Report

A very well constructed test-retest reliability study for a meat frequency questionnaire limited to the UK.

The major limitation is a lack of evaluation of g intake accuracy through comparison with a gold (if there is one) standard as noted in limitations.

Introduction:

The introduction  presents an appropriate justification and rationale for the research and is very well written and appropriately referenced.

Objectives:

Clear and concise.

Development process:

Thorough. I may have missed this detail, but how are mixed meats across animal groups covered eg a beef and lamb kebab

Line 90: The groupings seem to make sense, however how were the percentages for low vs high determined?

Please consider rewording: Two researchers worked independently on different animal groups throughout the whole development process; it is unclear as to whether they each did groups independently, worked on different groups?

Line 132: could paid Prolific recruitment have biased the sample? however demographics appear ok.

Line 142: Participants: "self-report to eat meat at least five times per week" Please consider whether this may have implications for reporting in occasional meat eaters.

Statistical anaylsis: congratulations on a transparent and pre-published plan.

Line 197. < or > ?

Intervention evaluation:

Completion times are fascinating. Thank you for the detail provided.

If possible, consider quotes in italics and/or demographic data in line with qualitative reporting

Discussion:

Line 347. Although this is a substantial assumption, I would tend to agree, and is adequately referenced.

Limitations:

The design doesn't account for intake of non-meat protein alternatives, which is a deliberate methodological approach. However the limitations of this chosen approach need to be considered, eg cannot account for changes in overall protein intake in intervention studies, which may be problematic.

Conclusions:

Appropriate and not overstated.

Reviewer 3 Report

Dear authors,

please find my comments in the attached pdf.

Round 2

Reviewer 3 Report

Thank you very much for addressing my concerns and answering my questions on your manuscript in such detail. I think the manuscript improved considerably. I just have one more note concerning the mean difference in total meat intake.

Table 4 and Line 280-282:

The numbers in table 4 are not totally in line with the text in line 280-282. In table 4, mean bias for total meat intake is 0.12. This is on the natural-log-scale and can be interpreted as percentage difference, which would be 12%, not 10%.  

Best wishes and all the best with your future research.

Author Response

Thank you very much for pointing out this mistake, we have amended the text on line 282 to read 12%. Thank you again for such constructive feedback. We're glad you feel the manuscript has improved, we also feel it has improved considerably.